# Information Competing Process for Learning Diversified Representations

**Jie Hu**[12], **Rongrong Ji**[123]*, **ShengChuan Zhang**[1], **Xiaoshuai Sun**[1],
**Qixiang Ye**[4], **Chia-Wen Lin**[5], **Qi Tian**[6].
[1]Media Analytics and Computing Lab, Department of Artificial Intelligence,
School of Informatics, Xiamen University.
[2]National Institute for Data Science in Health and Medicine, Xiamen University.
[3]Peng Cheng Laboratory. [4]University of Chinese Academy of Sciences.
[5]National Tsing Hua University. [6]Noah's Ark Lab, Huawei.

## Abstract

Learning representations with diversified information remains as an open problem. Towards learning diversified representations, a new approach, termed Information Competing Process (ICP), is proposed in this paper. Aiming to enrich the information carried by feature representations, ICP separates a representation into two parts with different mutual information constraints. The separated parts are forced to accomplish the downstream task independently in a competitive environment which prevents the two parts from learning what each other learned for the downstream task. Such competing parts are then combined synergistically to complete the task. By fusing representation parts learned competitively under different conditions, ICP facilitates obtaining diversified representations which contain rich information. Experiments on image classification and image reconstruction tasks demonstrate the great potential of ICP to learn discriminative and disentangled representations in both supervised and self-supervised learning settings. [1]

## 1 Introduction

Representation learning aims to make the learned feature representations more effective on extracting useful information from input for downstream tasks [4], which has been an active research topic in recent years and has become the foundation for many tasks [28, 8, 11, 15, 40, 20, 6]. Notably, a majority of works about representation learning have been studied from the viewpoint of mutual information constraint. For instance, the Information Bottleneck (IB) theory [38, 1] minimizes the information carried by representations to fit the target outputs, and the generative models such as $\beta$-VAE [13, 5] also rely on such information constraint to learn disentangled representations. Some other works [22, 3, 26, 14] reveal the advantages of maximizing the mutual information for learning discriminative representations. Despite the exciting progresses, learning diversified representations remains as an open problem. Diversified representations are learned with different constraints encouraging representation parts to extract various information from inputs, which results in powerful features to represent the inputs. In principle, a good representation learning approach is supposed to discriminate and disentangle the underlying explanatory factors hidden in the input [4]. However, this goal is hard to realize as the existing methods typically resort to only one type of information constraint. As a consequence, the information diversity of the learned representations is deteriorated.

In this paper we present a diversified representation learning scheme, termed Information Competing Process (ICP), which handles the above issues through a new information diversifying objective. First, the separated representation parts learned with different constraints are forced to accomplish the downstream task competitively. Then, the rival representation parts are combined to solve the downstream task synergistically. A novel solution is further proposed to optimize the new objective in both supervised and self-supervised learning settings.

We verify the effectiveness of the proposed ICP on both image classification and image reconstruction tasks, where neural networks are used as the feature extractors. In the supervised image classification task, we integrate ICP with four different network architectures (*i.e.*, VGG [34], GoogLeNet [35], ResNet [12], and DenseNet [16]) to demonstrate how the diversified representations boost classification accuracy. In the self-supervised image reconstruction task, we implement ICP with $\beta$-VAE [13] to investigate its ability of learning disentangled representations to reconstruct and manipulate the inputs. Empirical evaluations suggest that ICP fits finer labeled dataset and disentangles fine-grained semantic information for representations.

## 2 Related Work

**Representation Learning with Mutual Information.** Mutual information has been a powerful tool in representation learning for a long time. In the unsupervised setting, mutual information maximization is typically studied, which targets at adding specific information to the representation and forces the representation to be discriminative. For instance, the InfoMax principle [22, 3] advocates maximizing mutual information between the inputs and the representations, which forms the basis of independent component analysis [17]. Contrastive Predictive Coding [26] and Deep InfoMax [14] maximize mutual information between global and local representation pairs, or the input and global/local representation pairs.

In the supervised or self-supervised settings, mutual information minimization is commonly utilized. For instance, the Information Bottleneck (IB) theory [38] uses the information theoretic objective to constrain the mutual information between the input and the representation. IB was then introduced to deep neural networks [37, 33, 31], and Deep Variational Information Bottleneck (VIB) [1] was recently proposed to refine IB with a variational approximation. Another group of works in self-supervised setting adopt generative models to learn representations [19, 30], in which the mutual information plays an important role in learning disentangled representations. For instance, $\beta$-VAE [13] is a variant of Variation Auto-Encoder [19] that attempts to learn a disentangled representation by optimizing a heavily penalized objective with mutual information minimization. Recent works in [5, 18, 7] revise the objective of $\beta$-VAE by applying various constraints. One special case is InfoGAN [8], which maximizes the mutual information between representation and a factored Gaussian distribution. Besides, Mutual Information Neural Estimation [2] estimates the mutual information of continuous variables. Differing from the above schemes, the proposed ICP leverages both mutual information maximization and minimization to create competitive environment for learning diversified representations.

**Representation Collaboration.** The idea of collaborating neural representations can be found in Neural Expectation Maximization [10] and Tagger [9], which uses different representations to group and represent individual entities. The Competitive Collaboration [29] method is the most relevant to our work. It defines a three-player game with two competitors and a moderator, where the moderator takes the role of a critic and the two competitors collaborate to train the moderator. Unlike Competitive Collaboration, the proposed ICP enforces two (or more) representation parts to be complementary through different mutual information constraints for the same downstream task by a competitive environment, which endows the capability of learning more discriminative and disentangled representations.

## 3 Information Competing Process

The key idea of ICP is depicted in Fig. 1, in which different representation parts compete and collaborate with each other to diversify the information. In this section, we first unify supervised and self-supervised objectives for acheving the target tasks. Then, the information competing objective for learning diversified representations is proposed.

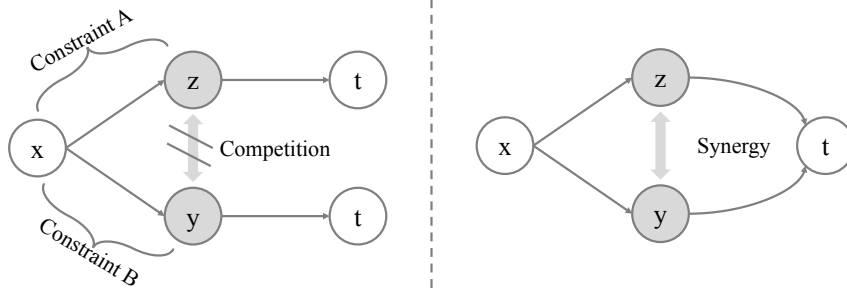

Figure 1: The proposed Information Competing Process. In the competitive step, the rival representation parts are forced to accomplish the downstream task solely by preventing both parts from knowing what each other learned under different constraints for the task. In the synergetic step, these representation parts are combined to complete the downstream task synthetically. ICP can be generalized to arbitrary number of constrained parts, and in this paper we make an example of two.

### 3.1 Unifying Supervised and Self-Supervised Objectives

The information constraining objective in supervised setting has the same form as that of self-supervised setting except the target outputs. We therefore unify these two objectives by using $t$ as the output of the downstream tasks. In supervised setting, $t$ represents the label of input $x$. In self-supervised setting, $t$ represents the input $x$ itself. This leads to the unified objective function linking the representation $r$ of input $x$ and target $t$ as:

$$\max \big[\mathcal{I}(r,t)\big], \tag{1}$$

where $\mathcal{I}(\cdot,\cdot)$ stands for the mutual information. This unified objective describes a constraint with the goal of maximizing the mutual information between the representation $r$ and the target $t$.

### 3.2 Separating and Diversifying Representations

To explicitly diversify the information on representations, we directly separate the representation $r$ into two parts $[z,y]$ with different constraints, and encourage representations to learn discrepant information from the input $x$. Specifically, we constrain the information capacity of representation part $z$ while increasing the information capacity of representation part $y$. To that effect, we have the following objective function:

$$\max \big[\mathcal{I}(r,t) + \alpha\mathcal{I}(y,x) - \beta\mathcal{I}(z,x)\big], \tag{2}$$

where $\alpha > 0$ and $\beta > 0$ are the regularization factors.

### 3.3 Competition of Representation Parts

To prevent any one of the representation parts from dominating the downstream task, we let $z$ and $y$ to accomplish the downstream task $t$ solely by utlizing the mutual information constraints $\mathcal{I}(z,t)$ and $\mathcal{I}(y,t)$. Additionally, for ensuring the representations catch diversified information through different constraints, ICP prevents $z$ and $y$ from knowing what each other learned for the downstream task, which is realized by enforcing $z$ and $y$ independent of each other. These constraints result in a competitive environment to enrich the information carried by representations. Correspondingly, the objective of ICP is concluded as:

$$\max \big[ \underbrace{\mathcal{I}(r,t)}_{\text{① Synergy}} + \underbrace{\alpha\mathcal{I}(y,x)}_{\text{② Maximization}} - \underbrace{\beta\mathcal{I}(z,x)}_{\text{③ Minimization}} + \underbrace{\mathcal{I}(z,t) + \mathcal{I}(y,t) - \gamma\mathcal{I}(z,y)}_{\text{④ Competition}}\big], \tag{3}$$

where $\gamma > 0$ is the regularization factor.

## 4 Optimizing the Objective of ICP

In this section, we derive a solution to optimize the objective of ICP. Although all terms of this objective have the same formulation that calculates the mutual information between two variables, they

need to be optimized using different methods due to their different aims. We therefore classify these terms as the mutual information minimization term $\mathcal{I}(z, x)$, the mutual information maximization term $\mathcal{I}(y, x)$, the inference terms $\mathcal{I}(z, t), \mathcal{I}(y, t), \mathcal{I}(r, t)$ and the predictability minimization term $\mathcal{I}(z, y)$ to find the solution.

## 4.1 Mutual Information Minimization Term

To minimize the mutual information between $x$ and $z$, we can find out a tractable upper bound for the intractable $\mathcal{I}(z, x)$. In the existing works [19, 1], $\mathcal{I}(z, x)$ is usually defined under the joint distribution of inputs and their encoding distribution, as it is the constraint between the inputs and the representations. Concretely, the formulation is derived as:

$$
\begin{aligned}
\mathcal{I}(z, x) = \int \int P(z, x) \log \frac{P(z, x)}{P(z)P(x)} dxdz &= \int \int P(z, x) \log \frac{P(z|x)}{P(z)} dxdz \\
&= \int \int P(z, x) \log P(z|x) dxdz - \int \int P(x|z)P(z) \log P(z) dxdz \\
&= \int \int P(z, x) \log P(z|x) dxdz - \int P(z) \log P(z) dz.
\end{aligned}
\tag{4}
$$

Let $Q(z)$ be a variational approximation of $P(z)$, we have:

$$
KL\big[P(z)||Q(z)\big] \geq 0 \Rightarrow \int P(z) \log P(z) dz \geq \int P(z) \log Q(z) dz.
\tag{5}
$$

According to Eq. 5, the trackable upper bound after applying the variational approximation is:

$$
\mathcal{I}(z, x) \leq \int \int P(z|x)P(x) \log \frac{P(z|x)}{Q(x)} dxdz = \mathbb{E}_{x \sim P(x)}\big[KL\big[P(z|x)||Q(z)\big]\big],
\tag{6}
$$

which enforces the extracted $z$ conditioned on $x$ to a predefined distribution $Q(z)$ such as a standard Gaussian distribution.

## 4.2 Mutual Information Maximization Term

To maximize the mutual information between $x$ and $y$, we deduce a tractable alternate for the intractable $\mathcal{I}(y, x)$. Specifically, like the above minimization term, the mutual information should also be defined as the joint distribution of inputs and their encoding distribution. As it is hard to derive a tractable lower bound for this term, we expand the mutual information as:

$$
\mathcal{I}(y, x) = \int \int P(y, x) \log \frac{P(y, x)}{P(y)P(x)} dxdy = KL\big[P(y|x)P(x)||P(y)P(x)\big].
\tag{7}
$$

Since Eq. 7 means that maximizing the mutual information is equal to enlarging the Kullback-Leibler (KL) divergence between distributions $P(y|x)P(x)$ and $P(y)P(x)$, and the maximization of KL divergence is divergent. We instead maximize the Jensen-Shannon (JS) divergence as an alternative which approximates the maximization of KL divergence but is convergent. As [25], a tractable variational estimation of JS divergence can be defined as:

$$
\begin{aligned}
JS\big[P(y|x)P(x)||P(y)P(x)\big] = \max \Big[ &\mathbb{E}_{(y,x) \sim P(y|x)P(x)}\big[\log D(y, x)\big] \\
&+ \mathbb{E}_{(\hat{y},x) \sim P(y)P(x)}\big[\log \big(1 - D(\hat{y}, x)\big)\big]\Big],
\end{aligned}
\tag{8}
$$

where $D$ is a discriminator that estimates the probability of the input pair, $(y, x)$ is the positive pair sampled from $P(y|x)P(x)$, and $(\hat{y}, x)$ is the negative pair sampled from $P(y)P(x)$. As $\hat{y}$ shoule be the representation conditioned on $x$, we disorganize $y$ in the positive pair $(x, y)$ to obtain the negative pair $(x, \hat{y})$.

## 4.3 Inference Term

The inference terms in Eq. 3 should be defined as the joint distribution of representation and the output distribution of downstream task solver. We take $\mathcal{I}(r, t)$ as an example, and $\mathcal{I}(z, t), \mathcal{I}(y, t)$

**Algorithm 1:** Optimization of Information Competing Process

---

**Input:** The source input $x$ with the downstream task target $t$, the prior distribution $Q(z)$, $Q(t|r)$, $Q(t|z)$ and $Q(t|y)$ for variational approximation, and the hyperparameters $\alpha, \beta, \gamma$.

**Output:** The learned representation extractor and downstream solver.

---

1 **while** *not Convergence* **do**
2     Optimize Eq. 8 and Eq. 16 for discriminator $D$ and predictor $H$;
3     *// Mutual Information Minimization Term:*
4     Replace $\mathcal{I}(z,x)$ in Eq. 3 with the tractable upper bound in Eq. 6;
5     *// Mutual Information Maximization Term:*
6     Replace $\mathcal{I}(y,x)$ in Eq. 3 with the tractable alternative in Eq. 8;
7     *// Inference Term:*
8     Replace $\mathcal{I}(z,t), \mathcal{I}(y,t), \mathcal{I}(r,t)$ in Eq. 3 with the tractable lower bound in Eq. 14;
9     *// Predictability Minimization Term:*
10     Replace $\mathcal{I}(z,y)$ in Eq. 3 with Eq. 16;
11     Optimize Eq. 3 while fixing the parameters of $D$ and $H$;
12 **end**

---

have the same formulation with $\mathcal{I}(r,t)$. We expand this mutual information term as:

$$
\begin{aligned}
\mathcal{I}(r,t) &= \int \int P(r,t) \log \frac{P(t|r)}{P(t)} dr dt \\
&= \int \int P(r,t) \log P(t|r) dt dr - \int P(t) \log P(t) dt \\
&= \int \int P(r,t) \log P(t|r) dt dr + \mathcal{H}(t) \\
&\geq \int \int P(t|r) p(r) \log P(t|r) dt dr,
\end{aligned}
\tag{9}
$$

where $\mathcal{H}(t) \geq 0$ is the information entropy of $t$. Let $Q(t|r)$ be a variational approximation of $P(t|r)$, we have:

$$
KL\big[P(t|r)||Q(t|r)\big] \geq 0 \Rightarrow \int P(t|r) \log P(t|r) dt \geq \int P(t|r) \log Q(t|r) dt. \tag{10}
$$

By applying the variational approximation, the trackable lower bound of the mutual information between $r$ and $t$ is:

$$
\mathcal{I}(r,t) \geq \int \int P(r,t) \log Q(t|r) dt dr. \tag{11}
$$

Based on the above formulation, we derive different objectives for the supervised and self-supervised settings in what follows.

**Supervised Setting.** In the supervised setting, $t$ represents the known target labels. By assuming that the representation $r$ is not dependent on the label $t$, *i.e.*, $P(r|x,t) = P(r|x)$, we have:

$$
P(x,r,t) = P(r|x,t)P(t|x)P(x) = P(r|x)P(t|x)P(x). \tag{12}
$$

Accordingly, the joint distribution of $r$ and $t$ can be written as:

$$
P(r,t) = \int P(x,r,t) dx = \int P(r|x)P(t|x)P(x) dx. \tag{13}
$$

Combining Eq. 11 with Eq. 13, we get the lower bound of the inference term in the supervised setting:

$$
\begin{aligned}
\mathcal{I}(r,t) &\geq \int \int \int P(x)P(r|x)P(t|x) \log Q(t|r) dt dr dx \\
&= \mathbb{E}_{x \sim P(x)} \Big[ \mathbb{E}_{r \sim P(r|x)} \big[ \int P(t|x) \log Q(t|r) dt \big] \Big].
\end{aligned}
\tag{14}
$$

Table 1: Classification error rates (%) on CIFAR-10 test set.

| | VGG16 [34] | GoogLeNet [35] | ResNet20 [12] | DenseNet40 [16] |
|---|---|---|---|---|
| Baseline | 6.67 | 4.92 | 7.63 | 5.83 |
| VIB [1] | $6.81^{\uparrow 0.14}$ | $5.09^{\uparrow 0.17}$ | $6.95^{\downarrow 0.68}$ | $5.72^{\downarrow 0.11}$ |
| DIM* [14] | $6.54^{\downarrow 0.13}$ | $4.65^{\downarrow 0.27}$ | $7.61^{\downarrow 0.02}$ | $6.15^{\uparrow 0.32}$ |
| $\text{VIB}_{\times 2}$ | $6.86^{\uparrow 0.19}$ | $4.88^{\downarrow 0.04}$ | $6.85^{\downarrow 0.78}$ | $6.36^{\uparrow 0.53}$ |
| $\text{DIM*}_{\times 2}$ | $7.24^{\uparrow 0.57}$ | $4.95^{\uparrow 0.03}$ | $7.46^{\downarrow 0.17}$ | $5.60^{\downarrow 0.23}$ |
| $\text{ICP-}_{\text{ALL}}$ | $6.97^{\uparrow 0.30}$ | $4.76^{\downarrow 0.16}$ | $6.47^{\downarrow 1.16}$ | $6.13^{\uparrow 0.30}$ |
| $\text{ICP-}_{\text{COM}}$ | $6.59^{\downarrow 0.08}$ | $4.67^{\downarrow 0.25}$ | $7.33^{\downarrow 0.30}$ | $5.63^{\downarrow 0.20}$ |
| **ICP** | $\mathbf{6.10}^{\downarrow 0.57}$ | $\mathbf{4.26}^{\downarrow 0.66}$ | $\mathbf{6.01}^{\downarrow 1.62}$ | $\mathbf{4.99}^{\downarrow 0.84}$ |

Table 2: Classification error rates (%) on CIFAR-100 test set.

| | VGG16 [34] | GoogLeNet [35] | ResNet20 [12] | DenseNet40 [16] |
|---|---|---|---|---|
| Baseline | 26.41 | 20.68 | 31.91 | 27.55 |
| VIB [1] | $26.56^{\uparrow 0.15}$ | $20.93^{\uparrow 0.25}$ | $30.84^{\downarrow 1.07}$ | $26.37^{\downarrow 1.18}$ |
| DIM* [14] | $26.74^{\uparrow 0.33}$ | $20.94^{\uparrow 0.26}$ | $32.62^{\uparrow 0.71}$ | $27.51^{\downarrow 0.04}$ |
| $\text{VIB}_{\times 2}$ | $26.08^{\downarrow 0.33}$ | $22.09^{\uparrow 1.41}$ | $29.74^{\downarrow 2.17}$ | $29.33^{\uparrow 1.78}$ |
| $\text{DIM*}_{\times 2}$ | $25.72^{\downarrow 0.69}$ | $21.74^{\uparrow 1.06}$ | $30.16^{\downarrow 1.75}$ | $27.15^{\downarrow 0.40}$ |
| $\text{ICP-}_{\text{ALL}}$ | $26.73^{\uparrow 0.32}$ | $20.90^{\uparrow 0.22}$ | $28.35^{\downarrow 3.56}$ | $27.51^{\downarrow 0.04}$ |
| $\text{ICP-}_{\text{COM}}$ | $26.37^{\downarrow 0.04}$ | $20.81^{\uparrow 0.13}$ | $32.76^{\uparrow 0.85}$ | $26.85^{\downarrow 0.70}$ |
| **ICP** | $\mathbf{24.54}^{\downarrow 1.87}$ | $\mathbf{18.55}^{\downarrow 2.13}$ | $\mathbf{28.13}^{\downarrow 3.78}$ | $\mathbf{24.52}^{\downarrow 3.03}$ |

Since the conditional probability $P(t|x)$ represents the distribution of labels in the supervised setting, Eq. 14 is actually the cross entropy loss for classification.

**Self-supervised Setting.** In the self-supervised setting, $t$ is the input $x$ itself. Therefore, Eq 11 can be directly derived as:

$$\mathcal{I}(r,x) \geq \int \int P(r|x)P(x) \log Q(x|r) dx dt = \mathbb{E}_{x \sim P(x)}\Big[\mathbb{E}_{r \sim P(r|x)}\big[\log Q(x|r)\big]\Big]. \qquad (15)$$

Assuming $Q(t|r)$ as a Gaussian distribution, Eq. 15 can be expanded as the L2 reconstruction loss for the input $x$.

### 4.4 Predictability Minimization Term

To diversify the information and prevent the dominance of one representation part, we constrain the mutual information between $z$ and $y$, which equals to make $z$ and $y$ be independent with each other. Inspired by [32], we introduce a predictor $H$ to fulfill this goal. Concretely, we let $H$ predict $y$ conditioned on $z$, and prevent the extractor from producing $z$ which can predict $y$. The same operation is conducted on $y$ to $z$. The corresponding objective is:

$$\min \max \Big[\mathbb{E}_{z \sim P(z|x)}\big[H(y|z)\big] + \mathbb{E}_{y \sim P(y|x)}\big[H(z|y)\big]\Big]. \qquad (16)$$

So far, we have all the tractable bounds and alternatives for optimizing the information diversifying objective of ICP. The optimization process is summarized in Alg. 1.

## 5 Experiments

In experiments, all the probabilistic feature extractors, task solvers, predictor and discriminator are implemented by neural networks. We suppose $Q(z), Q(t|r), Q(t|z), Q(t|y)$ are standard Gaussian distributions and use reparameterization trick by following VAE [19]. The objectives are differentiable and trained using backpropagation. In the classification task (supervised setting), we use one fully-connected layer as classifier. In the reconstruction task (self-supervised setting), multiple deconvolution layers are used as the decoder to reconstruct the inputs. The implementation details and the experimental logs are all avaliable at our source code page.

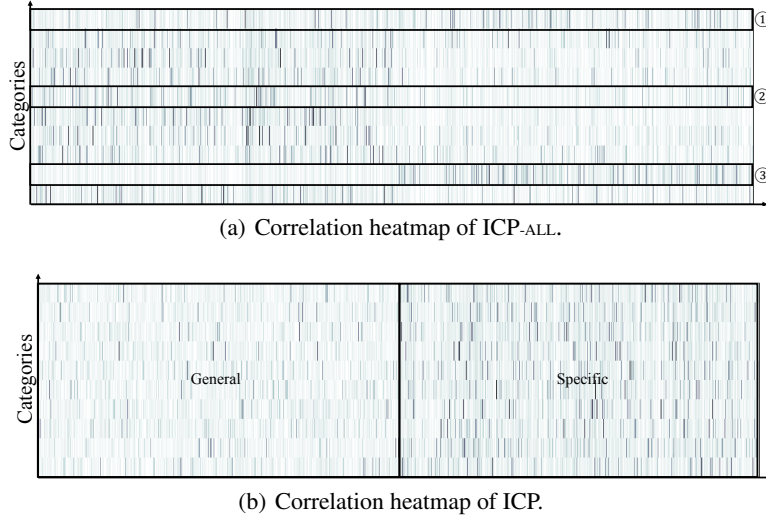

(a) Correlation heatmap of ICP-ALL.

(b) Correlation heatmap of ICP.

Figure 2: Heatmaps of the correlation between categories and the dimension of representations of VGGNet on CIFAR-10. The horizontal axis denotes the dimension of representations, and the vertical axis denotes the categories. Darker color denotes higher correlation.

## 5.1 Supervised Setting: Classification Tasks

### 5.1.1 Datasets

CIFAR-10 and CIFAR-100 [21] are used to evaluate the performance of ICP in the image classification task. These datasets contain natural images belonging to 10 and 100 classes respectively. CIFAR-100 comes with finer labels than CIFAR-10. The raw images are with $32\times32$ pixels and we normalize them using the channel means and standard deviations. Standard data augmentation by random cropping and mirroring is applied to the training set.

### 5.1.2 Classification Performance and Ablation Study

We utilize four architectures including VGGNet [34], GoogLeNet [35, 36], ResNet [12], and DenseNet [16] to test the general applicability of ICP and to study the diversified representations learned by ICP. We use the classification results of original network architectures as our baselines. The deep Variational Information Bottleneck (VIB) [1] and global version of Deep InfoMax with one additional mutual maximization term (DIM*) [14] are used as references, in which VIB is optimized by maximizing $\mathcal{I}(z,t) - \beta\mathcal{I}(z,x)$ , and DIM* is optimized by maximizing $\mathcal{I}(y,t) + \alpha\mathcal{I}(y,x)$. To make a fair comparison, we expand the representation dimension of both methods to the same size of ICP's (denoted as VIB$_{\times2}$, and DIM*$_{\times2}$). The VIB, DIM*, VIB$_{\times2}$ and DIM*$_{\times2}$ are the methods that only use one type of representation constraints in ICP, which can also be regarded as ablation study for ICP with single information constraint and without the information diversifying objective.

For further ablation study, we optimize ICP without all the information diversifying and competing constraints (*i.e.*, optimize Eq. 1), which is denoted as ICP-ALL. We also optimize ICP with the information diversifying objective but without the information competing objective (*i.e.*, optimize Eq. 2), which is denoted as ICP-COM.

The classification results on CIFAR-10 and CIFAR-100 are shown in Tables 1 and 2. We find that VIB, DIM*, VIB$_{\times2}$ and DIM*$_{\times2}$ achieve sub-optimal results due to the limited diversification of representations. ICP-ALL do not work well as the large model capacity overfits the training set, and ICP-COM fails because of the dominance of one type of representations. These results show that expanding models with sole constraint or removing one constraint from the objective decreases the performance. Only ICP generalizes to all these architectures and reports the best performance. In addition, the results on different datasets (*i.e.*, CIFAR-10 and CIFAR-100) suggest that ICP works better on the finer labeled dataset (*i.e.*, CIFAR-100). We attribute the success to the diversified representations that capture more detailed information of inputs.

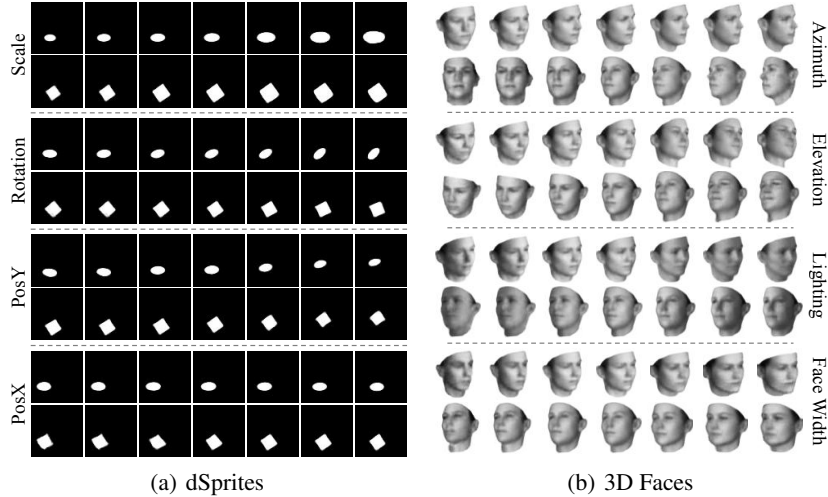

|       |       |
|-------|-------|
| (a) dSprites | (b) 3D Faces |

Figure 3: Qualitative disentanglement results of ICP on (a) dSprites and (b) 3D Faces datasets.

### 5.1.3 Interpretability of The Diversified Representations

To explain the intuitive idea and the superior results of ICP, we study the learned classification models to explore why ICP works and provide some insights about the interpretability of the learned representations. In the following, we make an example of VGGNet on CIFAR-10 and visualize the normalized absolute value of the classifier's weights. As shown in Fig. 2(a), the classification dependency is fused in ICP-ALL, which means combining two representations directly without any constraints does not diversify the representation. The first green bounding box shows that the classification relies on both parts. The second and the third green bounding boxes show that the classification relies more on the first part or the second part. On the contrary, as shown in Fig. 2(b), the classification dependency can be separated into two parts. As the mutual information minimization makes the representation carry more general information of input while the maximization makes the representation carry more specific information of input, a small number of dimensions are sufficient for inference (*i.e.*, the left bounding box of Fig. 2(b)), while a large number of dimensions are required for inference (*i.e.*, the right bounding box of Fig. 2(b)). This suggests that ICP learns diversified representations for classification.

## 5.2 Self-supervised Setting: Reconstruction

### 5.2.1 Datasets

We perform quantitative and qualitative disentanglement evaluations with the dataset of 2D shapes (dSprites) [24] and the dataset of synthetic 3D Faces [27]. The ground truth factors of dSprites are scale(6), rotation(40), posX(32) and posY(32). The ground truth factors of 3D Faces are azimuth(21), elevation(11) and lighting(11). Parentheses contain number of quantized values for each factor. The dSprites and 3D Faces contain 3 types of shapes and 50 identities, respectively, which are treated as noise during evaluation. The images of both datasets are reshaped to $64 \times 64$ pixels to compare with the baseline methods. We also evaluate the reconstruction and manipulation performance on more challenging CelebA [23] dataset which contains a large number of celebrity faces. The images are reshaped to $128 \times 128$ pixels for more detialed reconstruction instead of $64 \times 64$ pixels.

### 5.2.2 Quantitative Evaluation

We evaluate the disentanglement performance quantitatively by the Mutual Information Gap (MIG) score [7] with the 2D shapes (dSprites) [24] dataset and 3D Faces [27] dataset. MIG is a classifier-free information-theoretic disentanglement metric and is meaningful for any factorized latent distribution. As shown in Table 3, ICP achieves the state-of-the-art performance on the quantitative evaluation of disentanglement. We also conduct ablation studies as what we do in the supervised setting.

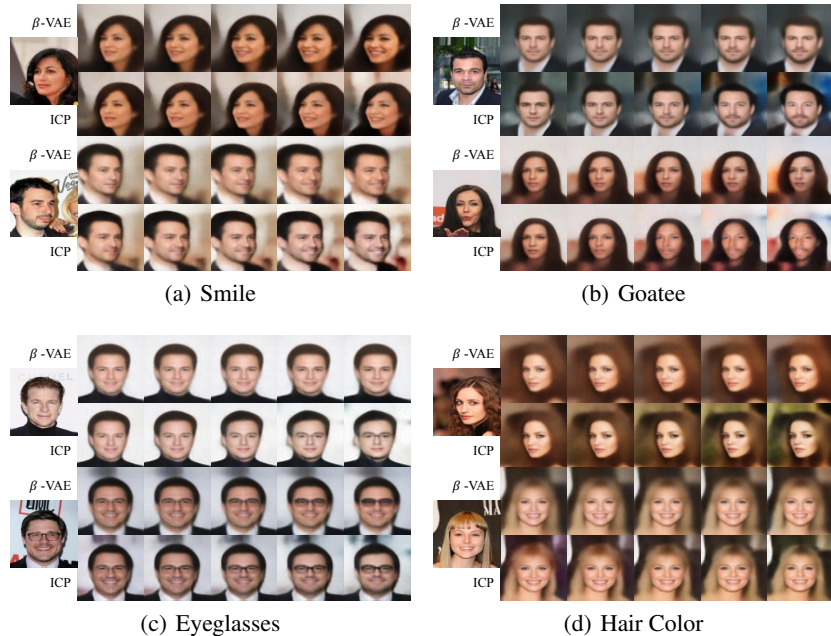

(a) Smile

(b) Goatee

(c) Eyeglasses

(d) Hair Color

Figure 4: Qualitative disentanglement results of $\beta$-VAE and ICP on CelebA. Each row represents a different seed image used to infer the representation.

From the results of ICP-ALL and ICP-COM, we find disentanglement performance decreases without the information diversifying and competing process. For the challenging CelebA [23] dataset, we evaluate the reconstruction performance via the average Mean Square Error (MSE) and the Structural Similarity Index (SSIM) [39]. The MSE of ICP is $8.5 * 10^{-3}$ compared with $9.2 * 10^{-3}$ of $\beta$-VAE [13] and the SSIM of ICP is $0.62$ compared with $0.60$ of $\beta$-VAE [13], which show ICP retains more information of input for reconstruction.

Table 3: MIG score of disentanglement.

|  | dSprites [24] | 3D Faces [27] |
|---|---|---|
| $\beta$-VAE [13] | 0.22 | 0.54 |
| $\beta$-TCVAE [7] | 0.38 | 0.62 |
| ICP-ALL | 0.33 | 0.26 |
| ICP-COM | 0.20 | 0.57 |
| **ICP** | **0.48** | **0.73** |

### 5.2.3 Qualitative Evaluation

For qualitative evaluation, we conduct the latent space traverse by traversing a single dimension of the learned representation over the range of [-3, 3] while keeping other dimensions fixed. We manually pick the dimensions which have semantic meaning related to human concepts from the reconstruction results. The qualitative disentanglement results are shown in Figs. 3 and 4. It can be seen that many fine-grained semantic attributes such as rotation on dSprites dataset, face width on 3D Face dataset and goatee on CelebA dataset are disentangled clearly by ICP with details.

## 6 Conclusion

We proposed a new approach named Information Competing Process (ICP) for learning diversified representations. To enrich the information carried by representations, ICP separates a representation into two parts with different mutual information constraints, and prevents both parts from knowing what each other learned for the downstream task. Such rival representations are then combined to accomplish the downstream task synthetically. Experiments demonstrated the great potential of ICP in both supervised and self-supervised settings. The nature behind the performance gain lies in that ICP has the ability to learn diversified representations, which provides fresh insights for the representation learning problem.

## Acknowledgments

This work is supported by the National Key R&D Program (No.2017YFC0113000, and No.2016YFB1001503), Nature Science Foundation of China (No.U1705262, No.61772443, No. 61802324, No.61572410 and No.61702136), and Nature Science Foundation of Fujian Province, China (No. 2017J01125 and No. 2018J01106).

## Footnotes

[1]Codes, models and experimental results are all available at `https://github.com/hujiecpp/InformationCompetingProcess/`

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
