[Reviews · NeurIPS 2019]

Reviewer 1



The main point of conduction to me is the main objective of this paper (not the objective they introduce). The authors write about “diversified representations”. Why is this useful? What does this mean? How is this different from disentangled representations? How did they show their own definition empirically. This main argument is hard for me to distill from the current version. That being said the paper is well written and easy to follow. Next I would like to ask for further clarifications on the objective that is been introduced, the method, itself. a) In section 3.3.2 , could you expand your explanations ? Why is (8) a bound on (7)? b) The optimisation seems very involved, can u say more about the hyper-parameter space you have been looking at/ sensitivities? c) Can you extend on the kind of networks that you have been using ? aka the backbone networks? d) What would happen if you let parts of your objective go? E.g. do you need term one Eq 3? e) What form does the discriminator network have? If I was to say that the classification performance is probably so good because the discriminator is pretty close to a what a classifier does, what would you say? Similarly like a GAN would have better FID scores than a VAE because the discriminator has such a similar functional form. And thus in that way you are somewhat cheating? In this submission, I would have enjoyed reading the code. The experiments seem competitive to other methods, even though error bars have not been provided. However again I can not find a clear hypothesis and respective experiments on what diversified means. In what context other than classification is it useful?

Reviewer 2



------ update after the rebuttal ------ the following points are revised from the discussion ------ 1/ In multiview learning, normally people could collect data from two conditionally independent resources or people could split the current existing data into two disjoint populations which creates multi-view in a cheap way. The way people split data into two disjoint populations could be thought of as minimising the mutual information between two "representations" of the same data. My point is that the authors shouldn't claim their work as totally independent/different from multi-view learning work in the rebuttal since IMO these two research lines are deeply connected. 2/ Maybe it is just my personal opinion. If the goal is to learn diversified vector representations, the paper needs to thoroughly justify the reason for using information bottleneck and also the whole variational inference, which was mentioned in my first review. To me, this paper threw variational autoencoders every where possible and didn't even both looking into the derivation and checking whether there were redundancy. From a probabilistic point of view, given x, y and z are naturally conditionally independent. Since the label is also presented in learning, which diminishes the conditional independence between y and z, the only thing we need to consider is to make sure that the mutual information between y and z is minimised, which could be implemented by a single adversarial classifier. That is the immediate thought I have when I was reading the title. With a certain Lipschitz constraint, people can prove error bounds on this issue. 3/ I couldn't figure out why I(r,t) term was necessary as there are two classifiers on y and z respectively, and the classification on the downstream tasks could rely on these two classifiers. ---- Variational inference is indeed interesting and also variational autoencoders are a huge win in the deep learning settings as SGD optimisations could be applied. However, we still need to carefully consider and justify why variational inference is necessary here. ---- ------end of the update ------- 1/ The necessity of the information bottleneck objective is doubtful. The goal of the information bottleneck objective is to learn "minimally sufficient statistics to transmit information in X to Y (which is T in this paper)". Given the goal here is to learn diversified representations of the input source, I don't see how the information bottleneck objective is being crucial here. 2/ Learning diversified objective through minimising the mutual information between two or among multiple pieces of information is not new. The performance gain one can get in multi-view learning or consensus maximisation is by ensuring that data representations (raw input representations or distributed representations learnt in neural networks) are conditionally independent given the correct label if they belong to the same class. Therefore, after learning, fusing multiple representations leads to improved performance compared to individual representation. This idea has been established around 30 years ago, and I recommend authors to at least refer to their papers in multi-view learning. 3/ If the main goal is to say that the proposed framework is capable of learning diversified representations and fusing representations gives better performance, then at least, a proper baseline should be, for a given baseline model, train two models with different initialisations and then take the ensemble of them when comparing performance. 4/ Lack of ablation study of the proposed objective. This relates back to my first point that the information bottleneck objective is not necessary and also the proposed paper didn't show why it was crucial to have it. Also, the objective function itself seems to be obese. For example, maximising I(z,t|\phi_z) + I(y,t|\phi_y) is a sufficient condition for maximising I(r,t|\phi) given that r is a concatenation of z and y. I hope the authors could critically evaluate their proposed method.

Reviewer 3



The authors examine the ability of mutual information constraints to diversify the information extracted to the latent representation. They study this by splitting the latent representation into two parts which are learned in competition using a mutual information constraint between them and opposing (min/max) mutual information constraints between the latent representations and the data. To train the model, the authors use a two-step competition and synergy training regime. The authors demonstrate the improved performance resulting from learning diversified representations in both the supervised and self-supervised setting. In the supervised setting, the authors show that the competing representations differ in their sensitivity to class-wise differences in the data. The difference is proposed to arise from the separation of high-level and low-level information learned by the two representations. In the self-supervised setting, the authors examine the role of the diversified representations in disentanglement and show increased performance compared to B-VAE and ICP. Major Comments: - The idea is well-motivated by the information bottleneck literature and the although the mutual information derivations present in this work are not novel (1), the role of diversified, competing representations in this context is not so well studied. Minor Comments: - an x-axis label stating which dimensions are from z and which are from y should be included. [1] Chen, T. Q., Li, X., Grosse, R. B., and Duvenaud, D. K. Isolating sources of disentanglement in variational autoencoders. In Advances in Neural Information Processing Systems, pp. 2610–2620, 2018 [2] Kim, H. and Mnih, A. Disentangling by factorising. In Proceedings of the 35th International Conference on Machine Learning, 2018.

[Author Response · NeurIPS 2019]

We thank all reviewers for their valuable comments, such as the novelty, well-motivated objective and promising results.

The code "ICP-pytorch" has been anonymously released on GitHub. We rebut key issues point-by-point as below.

**Response to R#2:**

We sincerely hope R#2 to raise the score. R#2 mainly criticized the missing of baseline and ablation study (indeed we

did have both), and misunderstood our flowchart to multi-view learning (indeed our flowchart has essential difference to

multi-view learning). We rebut these concerns below, and will clarify related issues in our paper.

**Q1: Similar idea with multi-view learning.** We rebut, with respect, that our flowchart is totally different from

multi-view learning. Our idea is to *distill* diverse representations with different constraints, while multi-view learning

focuses on the fusion of *predefined* features. Clearly, they reverse in procedures and diverse in mechanisms. To the best

of our knowledge, our work is the first attempt to explicitly model diversity in deep representation learning.

**Q2: Necessity of information bottleneck.** The information bottleneck objective (*i.e.*, Term ② in Eq.3r) is essential

for learning diversified representations as an information minimizing part, which results in representations that carry

general information. The necessity is also supported by the ablation study in Tab.1. To clarify, we reorder the objective

in Eq.3 as Eq.3r below. Term ① cannot be removed for ablation study solely, as it is the final target of our objective

which uses the constrained representation parts to complete the downstream task. Term ② (*i.e.*, information bottleneck

objective) and Term ③ (*i.e.*, information maximization objective) are different constraints for part $z$ and part $y$, which

force these parts to contain as more/less as possible information for the target task. $\mathcal{I}(z,t)$ and $\mathcal{I}(y,t)$ are used to

prevent any one of these features from dominating the task, and thus reduce the diversity of representations. Term ④ is

used to make $z$ and $y$ independent of each other. For qualitative evidence, please refer to our responses to Q3&Q4.

$$\max \Big[ \underbrace{\mathcal{I}(r,t)}_{\text{① Inference}} + \underbrace{\mathcal{I}(z,t) - \beta\mathcal{I}(z,x)}_{\text{② Minimize Information}} + \underbrace{\mathcal{I}(y,t) + \alpha\mathcal{I}(y,x)}_{\text{③ Maximize Information}} - \underbrace{\gamma\mathcal{I}(z,y)}_{\text{④ Independent}} \Big]. \quad (3r)$$

**Q3: The missing baseline.** Incorrect! Indeed, we did implement the mentioned baseline in Tab.1. Specifically, ICP-ALL

refers to the optimization using only Term ① of Eq.3r, which ensembles two models with different initializations as

introduced in L186. Overfitting occurs due to the large model capacity without constraints, making the performance of

this strategy sub-optimal for learning discriminative representations as shown in Tab.1.

**Q4: The lack of ablation study.** Incorrect! Indeed, we did provide necessary ablation studies in Tab.1. Specifically,

ICP-COM refers to optimization without $\mathcal{I}(z,t)$, $\mathcal{I}(y,t)$ and $\mathcal{I}(z,y)$, which means optimization without the competing

process. VIB refers to optimization that removes Term ③ of Eq.3r (*i.e.*, without information maximization constraint),

and DIM* refers to optimization that removes Term ② of Eq.3r (*i.e.*, without information bottleneck constraint), all

these achieve sub-optimal results due to the limited diversification of representations. VIB$_{\times 2}$/DIM*$_{\times 2}$ ensembles two

feature parts with the same constraints. These ablation studies show that expanding models with sole constraint or

removing one constraint from the objective decreases the performance.

**Response to R#1:**

**Q1: Meaning of diversified representations.** We apologize for the unclear definition in L23-27. Diversified represen-

tation learns representations with different mutual information constraints, which results in more powerful representation.

In contrast, disentangled representation, as a special case of our diversified representation, lacks diversity due to the

single mutual information minimization constraint.

**Q2: Confusions of Eq.8 and discriminator.** As KL divergence (Eq.7) has no upper bound, we use JS divergence

(Eq.8) as a substitution (not the bound). In Eq.8, the discriminator $D$ is used to maximize the JS divergence. $D$ only

distinguishes the positive pair $(x,y)$ and negative pair $(x,\hat{y})$, which is different from that of GAN.

**Q3: Explanation of Eq.3.** We reorder Eq.3 as Eq.3r. Term ① aims to fuse the feature parts for downstream tasks.

Term ② and Term ③ are different constraints together with Term ④ for diversifying the feature parts. All terms are

important for learning diversified representations and the ablation studies in Tab.1 support our claim.

**Q4: Implementation details and error bars.** We set small hyper-parameters (lr:0.01, $\gamma$:0.1, $\alpha$:0.1, $\beta$:0.01 in classifi-

cation; lr:1e-4, $\gamma$:1, $\alpha$:5, $\beta$:5 in reconstruction) to prevent gradient exploding. The error bars are listed below and ICP is

competitive. We will add these into our paper. If needed, please refer to our anonymous source code.

| CIFAR10 | VGG16 | GoogLeNet | ResNet20 | DenseNet40 | CIFAR100 | VGG16 | GoogLeNet | ResNet20 | DenseNet40 |
|---|---|---|---|---|---|---|---|---|---|
| Baseline | 6.67 | 4.92 | 7.63 | 5.83 | Baseline | 26.41 | 20.68 | 31.91 | 27.55 |
| ICP | $6.20 \pm 0.06$ | $4.48 \pm 0.11$ | $6.17 \pm 0.04$ | $5.27 \pm 0.13$ | ICP | $24.85 \pm 0.19$ | $19.06 \pm 0.13$ | $28.48 \pm 0.15$ | $25.48 \pm 0.22$ |

**Response to R#3:**

**Q1: Quantitative evaluation for disentanglement.** Following this suggestion, we have evaluated the MIG score

proposed in $\beta$-TCVAE with the same settings on dSprites and 3D Faces. The results of $\beta$-VAE are $0.21$ and $0.47$. ICP

are $0.22$ and $0.49$. We will add these results into our paper.

**Q2: More thorough explanation for the ablations.** We will add more in-depth analysis for the ablation studies in

Tab. 1. Due to the page limit, we cannot itemize them in rebuttal. In general, expanding models with sole constraint or

removing one constraint from our objective deteriorates the performance.

**Q3: Interpretability of deep representations.** Thanks for this inspiring suggestion. The interpretability of repre-

sentations learned by neural networks remains an open problem. Our experiment (Fig.2) provides insights about the

interpretability of learned representations. It shows that a small number of dimensions are sufficient for inference if

the representations contain general information, while a large number of dimensions are required for inference if the

representations contain specific information.

[Meta-Review · NeurIPS 2019]

This paper received two positive reviews and one quite negative review. So I read it carefully. The rebuttal has addressed the third point of Reviewer #2. I agree that maximizing I(z,t|\phi_z) + I(y,t|\phi_y) is sufficient for maximizing I(r,t|\phi), but this worry is more about redundancy than detriment. I personally don't find multiview so relevant here, although the technique of minimizing the mutual information between z and y is not new. Variational approximation is used extensively, and it looks effective. Most reviewers raised ablation study, but I think the paper has already conducted the most important ones: ICP-ALL and ICP-COM. Overall, the paper puts together some mutual information terms to learn diversified representations, facilitated by variational approximation. The experimental results look quite encouraging. So I would recommend it for publication. In the camera-ready, please address point e) of Reviewer #1, and incorporate all the modifications promised in the rebuttal.